# The Role of SNPs in the Pathogenesis of Idiopathic Central Precocious Puberty in Girls

**DOI:** 10.3390/children10030450

**Published:** 2023-02-25

**Authors:** Konstantina Toutoudaki, George Paltoglou, Dimitrios T. Papadimitriou, Anna Eleftheriades, Ermioni Tsarna, Panagiotis Christopoulos

**Affiliations:** 1Second Department of Obstetrics and Gynecology, “Aretaieion” Hospital, Faculty of Medicine, National and Kapodistrian University of Athens, 11528 Athens, Greece; 2Division of Endocrinology, Metabolism and Diabetes, ‘Aghia Sophia’ Children’s Hospital First Department of Pediatrics, Medical School, National and Kapodistrian University of Athens, 11527 Athens, Greece

**Keywords:** idiopathic central precocious puberty, single nucleotide polymorphisms, haplotypes, females, girls, genetics

## Abstract

The initiation of puberty is a crucial timepoint of development, with its disruptions being associated with multiple physical and psychological complications. Idiopathic Central Precocious Puberty (iCPP) has been correlated with Single-Nucleotide Polymorphisms (SNPs) of certain genes that are implicated in various steps of the process of pubertal onset. The aim of this review was to gather current knowledge on SNPs of genes associated with iCPP. We searched articles published on the PubMed, EMBASE and Google Scholar platforms and gathered current literature. KISS1, KISS1R, PLCB1, PRKCA, ITPR1, MKRN3, HPG axis genes, NPVF/NPFFR1, DLK1, KCNK9Q, LIN28B, PROK2R, IGF-1, IGF2, IGF-1R, IGF-2R, IGFBP-3, insulin, IRS-1, LEP/LEPR, PPARγ2, TAC3, TACR3, Estrogen receptors, CYP3A4 and CYP19A1 were studied for implication in the development of precocious puberty. SNPs discovered in genes KISS1, KISS1R, PLCB1, MKRN3, NPVF, LIN28B, PROK2R, IRS-1 TAC3, and CYP3A4 were significantly correlated with CPP, triggering or protecting from CPP. Haplotype (TTTA)13 in CYP19A1 was a significant contributor to CPP. Further investigation of the mechanisms implicated in the pathogenesis of CPP is required to broaden the understanding of these genes’ roles in CPP and possibly initiate targeted therapies.

## 1. Introduction

The timing of pubertal onset has been a topic that has not yet been fully elucidated, despite the extent of research conducted. It is well known that puberty constitutes an important time during one’s life since it is the era of physical and psychological changes that result in the acquisition of reproductive capacity [1,2]. Pubertal onset is affected by a variety of factors, including genetic, nutritional, and socioeconomic parameters [1,2]. Despite differences among the two sexes, the basic steps of pubertal development are quite similar among both males and females [3]. Puberty constitutes the result of Hypothalamic-Pituitary-Gonadal (HPG) axis re-activation [3]. In this setting, the maturation of GnRH-producing neurons leads to pulsatile hormone secretion, which, in turn, provokes gonadotropin release. Subsequently, the gonads (ovaries in females and testes in males) are activated, gaining the capacity of sex steroid and functional gamete production [3]. Many factors, such as kisspeptin, neurokinin B, dynorphin, and leptin regulate multiple steps of this process [4].

Clinically, the onset of puberty is defined as reaching Tanner and Marshall stage 2 in subjects of both sexes. In girls, thelarche is considered as the landmark of pubertal onset. Puberty is defined as precocious when it starts during a time earlier than that of 95% the individuals belonging to the same population. Thus, even though recent data show a decrease in the mean age of pubertal onset in Western countries, puberty in girls is typically considered precocious when it begins before the age of 6 in black girls and 7–8 in girls of different ethnicities, according to various sources [1,5].

Much research has focused on the possible implications of precocious puberty and menarche. Pubertal precocity is quite often responsible for low final height due to premature bone maturation [2]. Early menarche, on the other hand, has become an established risk factor for breast cancer development. A popular explanation for this correlation is the increase of exposure to estrogens and progesterone [6]. Accordingly, early menarche has been associated with obesity, hypertension, cardiovascular disease, and even multiple sclerosis without ignoring the proven psychosocial extensions, such as bullying victimization [1,6,7,8].

Precocious puberty can be distinguished as central (CPP) or peripheral (PPP) [2,9]. CPP, which is generally more frequent among females, results from hypothalamic–pituitary–gonadal axis activation [1,4,10]. This can be induced by pathology of the Central Nervous System (CNS) [1,4]. Nevertheless, it is idiopathic precocious puberty, which is the most common cause in females, representing approximately 90% of female-affecting cases, even though its incidence has declined because of the discovery of genes that influence the timing of puberty [1,4].

The human genome is known to be 99.5% identical among different individuals. Diversity in phenotypes occurs because of deviations in the remaining 0.5% along with epigenetic modifications [11]. Single-nucleotide polymorphisms (SNPs) are small modifications in the sequence of nucleotides, among individuals, that are partially responsible for the variety in their characteristics [12]. The exact definition of an SNP is that of a genome variation with a Minor Allele Frequency (MAF) greater than 1% [13]. According to a hypothesis, known as ‘common disease-common variant’, some SNPs might render subjects more susceptible to certain diseases [13]. The contribution of SNPs in the appearance of common diseases potentially lies in the cumulative effect of multiple variations that alter disease susceptibility [14]. In fact, the idea that SNPs could be used as genetic markers is not new [15].

### Aim

The aim of this study was to record the SNPs among different genes related to pubertal precocity and elucidate the mode of their functions. We attempted to gather elements concerning the effect of SNPs on precocious puberty phenotypes. We analyzed the mechanism of function of these genes and studied the role of the genetic deviations found to be statistically related to CPP in the creation of CPP phenotypes, wherever that was possible.

## 2. Materials and Methods

The study was conducted according to the instructions of the Preferred Reporting Items for Systematic Reviews and Meta-analyses (PRISMA) statement. The study protocol included the following steps: (i) original systematic literature search; and (ii) selection of appropriate studies to be included. KT and GP conducted a search of the appropriate literature simultaneously and independently. After thoroughly examining all titles and abstracts that appeared during the research, the two researchers selected articles that were relevant to the topic of interest and cooperated for a more detailed study of relevant literature. The time limit set during this process included studies that had been published until 25 May 2021.

The first database used was PubMed. The search was conducted using two groups of keywords. The Medical Subject Headings (MeSH) database was used for the identification of synonyms. These two groups were combined by the Boolean “AND” and the terms utilized within these search categories were combined by the Boolean “OR”. The full search strategy used for Pubmed was: (“Polymorphism, Genetic” [All Fields] AND (puberty, precocious [MeSH Terms]) OR (precocious puberty)” [All Fields] (Filters applied: Clinical Study). The initial search revealed 47 studies. After reading all titles and abstracts, 8 articles were excluded because of an evident lack of relevance. Subsequently, 8 more articles were excluded from this study since incompatibility with the aims of the study was detected. More precisely, articles containing male-only populations, research on genes based exclusively on animal models, and articles examining the role of specific SNPs on specific endocrinopathies, other than CPP, were excluded, leaving 31 articles eligible for review. Expansion of the research in Google Scholar and EMBASE revealed 4 more eligible articles that were included in our research. Thus, in total, 35 articles examining the association of SNPs and haplotypes with the occurrence of CPP were finally included in our study. For every gene with an established association to CPP, after genetic and statistical analysis, brief research for the mode of function was performed, followed by search of additional original articles indicating specific genetic polymorphisms’ correlation to CPP. Only original research articles were included in this systematic review.

The method of study selection is more thoroughly described in Figure 1.

## 3. Results

### Most Important Polymorphisms

KISS 1: The KISS1 gene is located in the long arm of the first chromosome (1q32) [16]. It consists of 3 exons. Parts of the second and third exon are translated to a precursor peptide of 145-aminoacids, which is then cleaved to a 54-aminoacid peptide. This peptide is, eventually, cleaved into 14 (108–121), 13 (109–121), or 10 (112–121) amino acid fragments, known as kisspeptins. All these fragments have the same affinity for the Kisspeptin Receptor (KISS1R/GPR54) since they possess a common C-terminal decapeptide. This allows attachment to GPR54 and eventual initiation of the pulsatile release of GnRH [16,17,18]. Kisspeptin administration in monkeys was followed by a surge of LH, indicating the important role of this peptide in the activation of the hypothalamic-pituitary-gonadal axis [19].

Two studies, one conducted in Chinese Han girls and one in Korean girls discovered a possible implication of the same SNP in the position 54650055 (G > T) with CPP [16,20]. Luan et al., discovered a statistically significant correlation of the SNP with early puberty, while Ko described a potentially protective role from the disease [16,20]. This polymorphism led to the substitution of proline by threonine in the 110th amino acid of the peptide, which is present in all forms of functional kisseptins [16,20]. According to Li et al., the SNPs rs1132506 and rs5780218 were significantly connected to CPP in Chinese Han girls, while the second SNP was also related to EFP (early and fast puberty) [17]. Rs1132506 was also one of the SNPs connected to CPP in Korean girls. Rs35128240 was the second SNP more common in patients than controls, while a novel polymorphism in the position 55648176, leading to a T > G, seemed to have a protective role [18]. No significant difference among the two groups was noted after the GnRH stimulation test [18]. The fact that many of these SNPs were situated in untranslated regions led researchers to believe they play a regulatory role in genetic expression [18]. Haplotypes GGA and GGGC-ACCC were found to play a protective role in the appearance of CPP [17,18].

KISS1R: KISS1R, formerly known as GPR54 is a gene of 3 kb, situated in the 19q13.3 region. It is the gene that codes a G-protein coupled receptor, consisting of 398 aminoacids and going through the cellular membrane 7 times. It is a peptide expressed in many tissues, including the placenta, pituitary, pancreas, and spinal cord. There seems to be a gradual increase in the levels of GPR54 in animal models during the pubertal years [19].

The sequencing of the GPR54 gene has revealed many polymorphisms. Luan et al., came across an SNP, located in the promoter region and more precisely in the position 855765, leading to a base change A/G, which proved to have a significant correlation to CPP [21]. Further analysis of the region surrounding the polymorphism indicated that in the presence of A, a c-fos regulating region was formed, whereas this region was absent when allele G appeared [21]. Another SNP, located in the fourth exon (position: 859955) resulted in the substitution of the 587th nucleotide, inducing an amino acid change from proline to histidine in the second extracellular loop. However, the finding did not have statistical significance and its functional effects remain under examination [21].

In another study, an SNP rs3050132 (c.1091 T/A), resulting in the substitution p.Leu364His and a novel intronic mutation (c.738 + 64 G/T) were found to have a statistically significant correlation with CPP, even though the hormonic profile and morphologic parameters were not different among those who carried them. The former polymorphism was also registered by researchers that studied pubertal precocity in Iranian CPP patients with a positive family history, but no controls were recruited in the study. Another notable finding was that the haplotype CAGTGTC might be contributing to the presentation of precocious puberty [22,23].

PLCB1, PRKCA, ITPR1: The KISS1/KISS1R pathway is completed by genes PLCB1, PRKCA, and ITPR1. The binding of kisspeptins to the KISS1 receptor is followed by activation of a phospholipase, named PLCβ1, which is encoded by PLCB1 gene. The phospholipase acts by hydrolysing phosphatidylinositol 4,5-biphosphate. Subsequently, inositol 1,4,5-triphosphate (IP3) and diacylglycerol (DAG), the most common second messengers, are produced. IP3 provokes Ca^2+^ mobilization after binding to the intracellular Ca^2+^ channel IP3 receptor that is encoded by ITPR1 gene. In its turn, DAG activates a kinase that is encoded by PRKCA. Even though ITPR1 and PRKCA cannot be used as genetic markers, due to their omnipresence in intracellular pathways, an SNP found in the PLCB1 was proved to play a statistically significant role in the pathogenesis of CPP. More precisely, rs708910, located in the 3′ UTR region of the PLCB1 gene was significantly more common in the patient group exclusively in recessive models (OR, 2.768; 95% CI, 1.305–5.872) [17].

MKRN3: MKRN3 is a gene that codes makorin ring finger protein 3. It is a gene that does not include introns. The gene’s function is not completely known. However, it seems to act as a ‘brake’ of pubertal onset by inhibiting factors that favor GnRH secretion via ubiquitination. It has been hypothesized that it might also participate in RNA and DNA binding and GnRH inhibition [24].

MKRN3 is a maternally imprinted gene situated in the 15th chromosome in the region deleted in Prader Willi Syndrome. This means that it is expressed only when inherited from the father [25].

MKRN3 seems to be an important contributor of pubertal regulation, as shown by the fact that it is the most common gene involved in CPP. Mutations that destruct MKRN3 architecture have been confirmed as causative factors of idiopathic CPP and are mostly found in Caucasians [26,27,28].

Abreu et al., discovered four mutations in MKRN3. Among them, 3 were frame-shift mutations that were associated with early interruption of the MKRN3 protein product. The fourth (p.Arg365Ser), a missence variant, was estimated to induce functional defects [24].

A novel heterozygous missense variant of MKRN3 (p. C340G) among two siblings, a girl with CPP and a boy with early puberty, was reported by a team of Greek researchers. A closer look at the genealogic tree of this family indicates an imprinted mode of inheritance of this potentially destructive variant [28]. An Italian study revealed 3 mutations, resulting in aminoacid changes (p.Arg328Cys, p.Pro160Cysfs*14, p.Cys410Ter) [26]. Studies conducted in Korean girls with CPP revealed some SNPs with lower frequency than in other ethnic groups and without apparent functional value, but the statistical significance of these findings was not counted since the studies did not include controls [29]. Another Korean cohort study only found one statistically significant connection of an SNP with CPP in males, but not in females [24]. rs12441827 was positively related to CPP only in males but not in females [30]. Pagani et al., examined polymorphisms in MKRN3 to find 3 SNPs, none of which seemed to have a significant association with CPP in girls. Additionally, the study did not reveal any mutations that are classically related to earlier age of pubertal onset [31].

DLK1: DLK1 is a gene coding a transmembrane protein that participates in adipose tissue and neurogenesis-related processes [26]. Similarly to MKRN3, it is reported as an inhibitor of pubertal processes [32] expressed in the hypothalamus [33]. It is a maternally imprinted gene situated in the 14q32.2 region (that consists of a number of genes that follow the imprinted mode of inheritance) [34]. It is known as the causative gene of Temple Syndrome, which consists of growth failure, dysmorphic features, hypotonia, truncal obesity, and CPP [35]. While the exact process through which DLK1 impacts pubertal onset is not known, there seems to be a complex pathway that affects kisspeptin neuron development, involving the NOTCH signaling pathway [33]. A correlation of DLK1 with CPP was established when a complex aberration of the gene (14-kb deletion, combined with a 269-bp duplication) was identified in a Brazilian family with multiple cases of CPP among female members [33]. Three loss-of-function mutations in the DLK1 gene (p.Gly199Alafs*11, p. Val271Cysfs*14, and p.Pro160Leufs*50) have been associated with CPP [35]. Interestingly, CPP was accompanied by endocrine disturbances, such as obesity, insulin resistance, type 2 diabetes, and hyperlipidemia [35]. In one case of familial CPP, two sisters presented with PCOS [35]. Thus, even though DLK1 is a gene with an established role in CPP appearance, in the presence of deletional mutations, Grandone et al., did not reveal any SNPs correlated with the early appearance of puberty among 60 girls with CPP [26,35].

KCNK9Q: KCNK9Q is a maternally imprinted gene that codes a potassium channel, which regulates neuronal function. Although this gene has been implicated with the age of menarche, it might not be considered as a common contributor to the genetic basis of CPP, since SNPs have not been detected among 60 CPP patients participating in an Italian study [26].

HPG axis genes: The regulation of pubertal development by hormones guides research towards the genes coding the compounds of the HPG axis. Mutations of GnRH receptor gene were proved responsible for Idiopathic Hypogonadotrophic Hypogonadism (IHH), indicating the possible role of different GnRH gene mutations in pubertal precocity [36]. Variants in genes GNRH1 and GNRHR genes have also been identified in a 6.5-year-old girl with ovarian-cyst-derived peripheral precocious puberty [37]. These genes are relatively common causes of hypogonadotrophic hypogonadism [37]. Even though the first results from small-scale research on GnRH gene were not fruitful [38], data collected by Zhao et al., indicated that gonadotropin levels are severely influenced by changes in the upstream gene regions [15]. There were many references on mutations in these regions, resulting in quantitative and qualitative changes in these hormones and, therefore, pubertal timing. During their study, they examined SNPs present in the regulatory regions of these genes among Chinese girls. The one found in GnRH (rs2321049) did not have statistical significance. As far as the LHβ gene is concerned, the GCAAA haplotype was positively related to pubertal abnormalities, whereas GCGGT and CCAAA had a protective role. Finally, concerning the FSHb gene, the SNP rs639403 (T/C) was the only polymorphism significantly related to CPP. Further analysis of this SNP revealed that it did not reside in the transcription factor binding region [15].

NPVF/NPFFR1: GnIH is a hormone that is present in avians and acts as a GnRH suppressor. It is secreted by hypothalamic neurons and exerts its actions by binding to a G-protein-coupled receptor found in the anterior pituitary [39,40]. In mammals, NPVF is considered to be the gene that produces the analogue of GnIH, that is, RFRP-3. To be precise, NPVF is translated into a precursor peptide that is then cleaved in RFRP-1, RFRP-2, and RFRP-3 [39,41]. In parallel, NPFFR1 is the gene coding the G-protein-coupled receptor, GPR-147, binding to RFRP-3. Even though researchers who studied polymorphisms in this gene expected that a destructive mutation would cancel its inhibitory role and therefore lead to CPP, the results of their study indicated the opposite. P.I71_K72, a mutation that resulted in a 3-nucleotide deletion and consequently isoleucine removal, was proved to be protective of CPP in a Brazilian study. The same research discovered 12 polymorphisms, 5 of which were missence, in the NPFFR1 gene, none of which were significantly linked to CPP or to idiopathic hypogonadotrophic hypogonadism [42].

PROK2R: This gene codes for a G-protein-coupled receptor on the membrane of GnRH neurons. Its stimulation leads to GnRH secretion. Thus, an upregulating variation is assumed to contribute to CPP, occurring in a very young age. An Italian study, conducted by examining female CPP patients with an age of pubertal onset lower than 6 years old, revealed 5 SNPs in PROK2R. Only one (rs3746682, c.585G > C) was found to have a statistically higher minor allele frequency (MAF) (=0.84) than the one given by GnomAd. It is a synonymous polymorphism (p. Thr195=) that may not have any functional effects [43].

LIN28B: Even though the exact role of LIN28B is not known, it is often examined as a contributor to CPP. Many studies refer to it as a regulator of early menarche [44]. Other studies refer to it as a regulator of miRNA processing [45,46]. Gain-of-function mutations in this gene are known to lead to delayed puberty, while loss-of-function mutations intrigue precocious puberty [45]. A study concerning Taiwanese girls found a significant connection of CPP with the genotype frequencies rs221634 (*p* = 0.01) and rs314276 (*p* = 0.02). For rs221634 only the dominant model was significantly different. For rs314276, however, both dominant and recessive models were shown to have significant differences among patients and controls. This polymorphism also indicated a clear tendency to higher standard deviation scores (SDS) of weight and body mass index (BMI) in CPP patients when CPP patients were homozygous with the C allele, which was recorded as the dominant trait [47]. The rs314276 polymorphism was indicated as the sole polymorphism to reach genome-wide statistical significance after a genome-wide association analysis [46]. It was proven to be associated with a mean 0.22 years earlier age at menarche, but also earlier breast development and lower final adult height [46]. The CC genotype of rs7759938 and the AA genotype of rs314280, shaped by the combination of minor alleles, significantly lowered the risk of CPP in Chinese Han girls [48]. The first is a polymorphism in the 3′ UTR, whereas the second resides in the intron [48,49]. However, neither rs314276 nor rs314280 had a significant association with CPP in Korean girls [30]. These results are in accordance with those of a Taiwanese study that also failed to prove a significant link between rs314280 and CPP [49]. An AC haplotype, situated in the 5′ region of the second intron of the LIN28B gene, that could be reconstituted by rs4946651 and rs369065 had significantly lower incidence among Korean CPP patients [45].

IGF-1, IGF2, IGF-1R, IGF-2R, IGFBP-3, insulin: The IGF pathway consists of IGF-1, IGF-2, IGF receptors (IGFR-1, IGFR-2) and 6 IGF-binding proteins (IGFBPs) that interfere with IGF transfer and protect from protein degradation in the periphery [50]. IGF-1 is known to play a leading role in growth and tissue repair [51]. Chang et al., conducted a study that confirmed IGF-1 and IGFBP-3 levels in relation with commonly assessed growth parameters, such as bone age and LH for the former, and bone age, BA/CA ratio, and FSH for the latter. Even though SNPs of these genes showed significant differences among patients and controls with different demographic and pathological features, no significant connection to timing of CPP was found. However, some SNPs combinations had significant effect on parameters, including age of pubarche in the CPP patients [IGF-2(6093) + IGFBP-3] and IGF levels [IGF-1(1770) + IGFBP-3] (*p* = 0.038), [IGF-1(6093) + IGFBP-3] (*p* = 0.013), and [IGF-2(3580) + IGFBP-3] (*p* = 0.036) [52].

IRS-1: Insulin receptor substrate 1 is implicated in insulin metabolic functions. The hormone binds to its transmembranic receptor (IR), which in turn is activated by autophosphorylation and phosphorylates IRS. Thus, after a long molecular pathway RNA, protein and glycogen synthesis are induced [53]. IRS-1 972R was significantly correlated with the timing of pubertal onset, with a notably higher frequency in the control group [54].

PPARγ2: The PPARγ2 genetic polymorphisms were examined considering their relation to the IGF pathway. The Pro12Ala SNP, however, did not seem to exert any effect on the timing of adrenarche, even though the Ala12 variant was associated with lower growth potential [55].

TAC3 and TACR3: TAC3 is the gene responsible for preprotachykinin B production, which is then cleaved to neurokinin B (NKB). TACR3 codes the NK3R receptor, belonging in the rhodopsin family of receptors, which is coupled with a Gq-protein. A Turkish study examining the genetic base of familiar IHH highlighted the contribution of NKB and NK3R in regulation of the reproductive system [36]. The NKB binding to its receptor results in a calcium influx towards the cell [36]. NKB is produced in the arcuate nucleus. A rare variant (p.A63P) that was found in the coding region of proneurokinin B of a Brazilian female patient with CPP was not proven to play a significant role in pubertal timing [56]. The same SNP was reported as a statistically significant contributor to CPP in a Chinese study [57]. This is a variant that results from a substitution of G by C in nucleotide 187. Assumptions that this variant was devastating for the final product were not proven.

Estrogen receptors: Estrogenic stimulation of hormone-sensitive tissues constitutes the final step in the molecular cascade of events, leading to pubertal development. Estrogens exert their actions by binding to intracellular estrogenic receptors in their target tissues. Two estrogen receptors, ERα and ERβ, have been discovered. The function of these receptors might be affected by their genetic structure, as shown by Lee et al. [58]. Their study revealed a significant connection between SNP 15219480G/A in the ERα gene, inducing a substitution of glycine by serine in the 145th position, and CPP. GnRH stimulation testing in patients carrying these polymorphisms resulted in significantly higher levels of LH.

Xbal (351 A/G) and Pvull (397 T/C) are among the most-investigated polymorphisms of the ERα gene. These are intronic sequences, residing in intron 1 [59]. Stavrou et al., proposed that these polymorphisms could participate in defining the timing of menarche by delaying it [60]. Nevertheless, several other studies failed to reveal a correlation between these SNPs and PP or premature thelarche [58,59,61]. A Korean study conducted by Lee et al., rejected a possible association between these polymorphisms and PP in the local population [59]. A notable finding was that Xbal carriers were prone to a greater Tanner stage [62].

CYP3A4: Cytochrome P450 (CYP) enzymes are proteins that contain heme and participate, among others, in metabolic pathways of various hormones [63]. CYP3A4, a member of the cytochrome P450 superfamily, is the predominant isoform of CYP enzymes that is present in the majority of human livers [64]. A high-activity variant of CYP3A4, by the name of CYP3A4*1B, was found to be a significant contributor to pubertal onset [65]. However, the correlation was not confirmed in the study conducted by Xin et al. More precisely, CYP3A4*1B, CYP3A4*4, CYP3A4*5, and CYP3A4*6 variations did not have a significant effect in the appearance of CPP when 176 affected Chinese patients were examined, even though an association with pubertal onset was hypothesized [66].

CYP19A1: This gene encodes aromatase, an enzyme that is responsible for androgen to estrogen conversion. Thus, aromatase activity might affect the levels of estrogens. Further genetic analysis revealed polymorphisms in the repeat number of the (TTTA) sequence in intron 4. More precisely, the number of repeats varied between 7 and 13. (TTTA)_13_ was possibly linked to CPP and higher concentration of estrogens, whereas SNPs that were found did not have any statistical significance [67].

LEP/LEPR: LEP codes leptin, while LEPR codes leptin receptor [68]. A leptin receptor is a protein with 6 isoforms, produced after alternative splicing of the responsible gene [69,70]. Binding of leptin to its receptor initiates a complex, multistep JAK2-STAT3-mediated pathway that gives rise to processes implicated in energy homeostasis and reproduction [70,71]. Leptin seems to have mostly a regulatory role in the onset of puberty, even though its exact contribution in the activation of the HPG axis is not well defined [71]. Gueorguiev et al., reported that leptin has a permissive role in pubertal onset, but also in conserving reproductive capacity afterwards [71]. In a study by Su et al., even though leptin levels were clearly higher in a group of CPP patients, no SNP studied in these genes was proven to significantly affect leptin levels [68].

The most important SNPs and haplotypes that have been linked to CPP, in an either triggering or protective manner, are shown in Table 1, Table 2 and Table 3 below.

## 4. Discussion

Given the physiologic functions of the abovementioned genes, mutations leading to gain of function in the KISS1/KISS1R pathway genes along with loss of function mutations in MKRN3 and DLK1 were extensively studied for their potential causative effects in the presentation of CPP.

As discussed above, numerous polymorphisms have been found to play a statistically significant role in the pathogenesis of CPP. Polymorphisms in the KISS1/KISS1R pathway, the MKRN3, as well as the genes of the HPG axis are among the most commonly involved as statistically significant contributors of CPP, having either triggering or protective functions (Table 1 and Table 2). Haplotypes have also been investigated (Table 3). Despite the extent of research, there are still no adequate findings to thoroughly explain the pathogenesis of familial CPP. Even today, the exact genes implicated have not been defined. Additionally, the mechanisms via which the genetic deviations interfere in pubertal timing are indeterminate and possibly heterogenous. Mutations leading to functional defects, polymorphisms in non-translatable and regulatory regions, and even gene-to-gene interaction have been linked or hypothesized as participants in the creation of the CPP phenotype [57].

MKRN3, DLK1, KCNK9, and LIN28b are recognized as imprinted genes, and have all been implicated in CPP investigation. This means that gene expression is epigenetically permitted or prohibited, depending on parental origin. Epigenetic modifications utilise DNA methylation and demethylation mechanisms to modify genetic expression in a dynamic pattern throughout mammalian development [72,73,74]. The role of epigenetics in the pathogenesis of various human diseases has been thoroughly examined [74]. Thus, epigenetics and imprinting have recently been put under investigation regarding the physiology of pubertal onset and pathophysiology of its deviations [75]. According to Roberts et al., imprinted genes might significantly contribute to the regulation of pubertal timing [75]. Bessa et al., reported that normal puberty comes with methylation of certain regions of the human genome, especially in chromosome X [32]. The same study reported that girls with CPP had an even higher rate of methylated CpG regions [32]. However, there was no detection of methylation deviations in MKRN3 and DLK1 genes [32].

Another keystone of our study is the great number of SNPs with significant correlations with CPP that are situated in regions that are not coded during the translation process. Interestingly, this is a common finding, as genome-Wide association studies (GWAS) have indicated that most SNPs with some correlations with complex diseases reside in untranslated regions of the human genome [14,76]. This phenomenon is partially explained by the hypothesis that these regions might be untranslated but still be functional, exerting regulatory functions of the genes in proximity [14,76].

Another notable finding is that the frequency of diverse polymorphisms is different among patients with different ethnic backgrounds. This reinforces the need for extended research and sequencing among females all over the world to discover more related polymorphisms and clarify the pathophysiology of pubertal precocity.

A final point of interest is the deviation in CPP incidence among the two sexes. As discussed above, ICPP is known to be significantly more common among females [77,78,79]. This might be a result of what is known as sexual dimorphism. In fact, both normal puberty and its disorders are sexually dimorphic [80]. In this setting, kisspeptin, a peptide that is pivotal in pubertal development, constitutes a commonly described example of sexual dimorphism expression [80]. Kisspeptin levels were shown to be more elevated in adult females compared with males [81].

## 5. Conclusions

In conclusion, the genetic basis of CPP has been discussed extensively in the past. Multiple genetic polymorphisms among diverse genes have been studied, with some genetic polymorphisms appearing as correlated with precocious puberty phenotypes. Evaluation of family history in individuals affected by CPP is a prerequisite, especially when idiopathic precocious puberty is suspected. In this direction, some even suggest systematic genetic testing in the presence of a positive family history [28]. In any case, investigation of the genetic base of early and precocious puberty might take research to new lengths by establishing genetic markers for early diagnosis and aiding the development of targeted pharmacogenetic therapies [66,82].

## Figures and Tables

**Figure 1 children-10-00450-f001:**
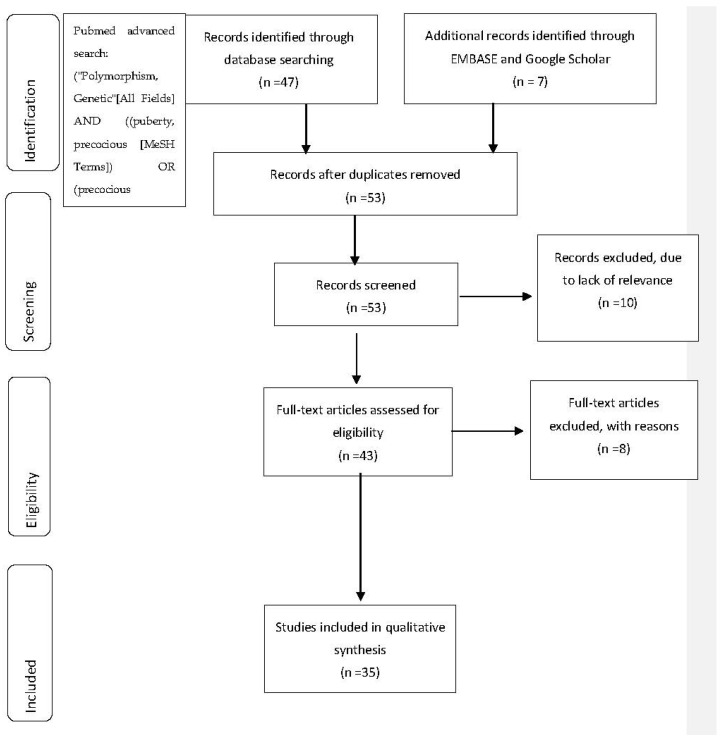
Methods of study selection.

**Table 1 children-10-00450-t001:** SNPs found to have statistically significant triggering correlations with CPP 1.

Gene	Polymorphism Position	Major/Minor Allele	DbSNP ID	Location	Expression	Study	Country	Comments
KISS1	54650055 *	G/T	-	Exon 3	110th aminoacid p.P110T	Luan et al. [20] Ko et al. [16]	China (Han) Korea	Statistical correlation with CPP (*p*-value = 0.025) Significantly protective role (*p*-value = 0.022)
	55648184	C/G	rs1132506	Exon 3	3′-UTR	Li et al. [17] Rhie et al. [18]	China (Han) Korea	Significant association with early puberty in A, D ^a^ More common among patients than controls (*p*-value = 0.017)
	204196482	A/-	rs5780218		5′-UTR	Li et al. [17]	China (Han)	Significant association with early puberty in A, D, and R ^a^
	55648186	-/T	rs35128240	Exon 3	3′-UTR	Rhie et al. [18]	Korea	More common among patients than controls (*p*-value = 0.044)
KISS1R	855765	A/G	-	Promoter region	5′-UTR	Luan et al. [21]	China (Han)	Statistical correlation with CPP (*p*-value = 0.037)
	c.1091	T/A	rs3050132	Exon 5	Codon 364 p.Leu364His	Ghaemi et al. [22] Oh Y.J. et al. [23]	Iran Korea	p.Leu364His (no statistical research) was present in 64% of all cases Significantly more common in D ^a^ Significantly more frequent among CPP patients (*p* = 0.031)
	c.738 + 64	G/T	rs350131	Intron 4		Oh Y.J. et al. [23]	Korea	CPP more common in R ^a^ (*p* = 0.006). Significantly higher allele frequencies in CPP patients than in controls (*p*-value = 0.023)
PLCB1	8883556	A/G	rs708910		3′ UTR	Li et al. [17]	China (Han)	Significantly associated with early puberty in R ^a^
MKRN3	c.1018	T/G	-	C3HC4 Ring motif	p.C340G	Settas et al. [28]	Greece	The missense variant, a probable damaging mutation, was present in both siblings affected by CPP, following an imprinted mode of inheritance.
FSHβ	30230078	T/C	rs639403		Untranslated region, not regulatory	Zhao et al. [15]	China	Weak correlation with CPP (*p*-value = 0.025)
LIN28B	chr6:105080213	A/T	rs221634		3′ UTR	Chen et al. [47]	Taiwan	Significant correlation with CPP in D ^a^
	6:104960124	A/C	rs314276		3′ UTR	Chen et al. [47] Hu et al. [48] Ong et al. [46]	Taiwan United Kingdom	Significant correlation with CPP in D, R ^a^ No statistical significance after adjusting for multiple testing Association with earlier breast development, 0.12-years-earlier menarche and shorter final height.
		A/C	rs4946651/rs369065		5′ region to intron 2	Park et al. [45]	Korea	
TAC3		G/C			A63P mutant	Xin et al. [57]	China	Significantly more common among patients compared with controls (*p* = 0.024)

Data were crosschecked from articles and NIH on the NCBI platform; ^a^: models of inheritance in which statistical significance was proved, D: dominant models, R: recessive models, A: additive models, * Polymporphism that has been reported as both triggering and protective, among diverse studies.

**Table 2 children-10-00450-t002:** SNPs found to have statistically significant protective correlations with CPP.

Gene	Polymorphism Position	Major/Minor Allele	DbSNP ID	Location	Expression	Study	Country	Comments
KISS1	54650055 *	G/T	-	Exon 3	110th aminoacid p.P110T	Ko et al. [16] Luan et al. [20]	China (Han) Korea	Significantly protective role (*p*-value = 0.022) Statistical correlation with CPP (*p*-value = 0.025)
	55648176	T/G	-	Exon 3		Rhie et al. [18]	Korea	Significantly protective role (*p*-value= 0.030)
NPVF	7:25226954 (GRCh38) 7:25266573 (GRCh37)	TAA>-	rs3216928	Exon 2	c.212_214del	Lima et al. [42]	Brazil	Significanly lower probability of CPP (OR = 0.33; 95% CI = 0.08–0.88)
LIN28B	104931079	T/C	rs7759938 *		3′ UTR	Hu et al. [48]	China (Han)	Decreased risk of CPP with the CC genotype. Significant association to CPP in both A ^a^ and R ^a^ (after adjusting for multiple testing)
	104952962	G/A C/T	rs314280 *		Intron	Hu et al. [48] Chou et al. [49]	China (Han) Taiwan	Significant association to CPP in both A, R ^a^. (after adjusting for multiple testing). Minor allele phenotype (AA) was protective from CPP Not statistically significant correlation (*p*-value = 0.1045)
		A/C	rs4946651/rs369065	Intron 2	5′ region to intron 2	Park et al. [45]	Korea	Significantly less common among CPP patients. Non-A/C haplotypes associated with CPP.
IRS-1			IRS-1 R972			Xin et al. [54]	China	Significantly more common among controls (2.6%) compared with patients (0.6%) (*p*-value = 0.043)

Platform ^a^: models of inheritance in which statistical significance was proved, D: dominant models, R: recessive models, A: additive models, * Polymporphism that has been reported as both triggering and protective, among diverse studies.

**Table 3 children-10-00450-t003:** Haplotypes with statistically significant correlations with CPP.

Gene	Haplotype	Study	*p*-Value
KISS1	GGA *	Li et al. [17]	0.005
	GGGC–ACCC *	Rhie et al. [18]	0.024
KISS1R	CAGTGTC	Oh YJ et al. [23]	0.042
LHβ	GCAAA	Zhao et al. [15]	7.77 × 10^−4^
	GCGGT *	Zhao et al. [15]	4.67 × 10^−6^
	CCAAA *	Zhao et al. [15]	1.21 × 10^−4^
CYP19A1	(TTTA)_13_	Lee et al. [67]	0.033

*: Protective role from precocious puberty.

## Data Availability

Not applicable.

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
