# Peer review of "The Role of SNPs in the Pathogenesis of Idiopathic Central Precocious Puberty in Girls"

_children, 2023, doi:10.3390/children10030450_

Round 1

Reviewer 1 Report

Konstantina Toutoudaki  and colleagues are presenting an interesting and complete review regarding Single Nucleotide Polimorfism variations in the pathogenesis of Central Precocious Puberty in females.  The review is of clinical interest since this conditions disproportionately  affects  girls  compared  to  boys and genotype – phenotype correlations in the field is still a matter of research.

However, The Authors should consider the following modifications:

11)     The most common genes known to be associated in the conditions (MRKNS, DLK1 And KISS-KISS1R) should be stressed further.

Mutation in KISS1-KISS1 r should be highlighted as gain of function, mutation in MKRN3  protein that physiologically has a role in repressing pubertal  initiation should be highlighted as loss of function, while mutations in DLK1 gene, a paternally expressed gene very common in the adrenal, pituitary and ovaric tissue, should be described as loss of function.

[see reference: Roberts SA, Kaiser UB. GENETICS IN ENDOCRINOLOGY: Genetic etiologies of central precocious puberty and the role of imprinted genes. Eur J Endocrinol. 2020 Oct;183(4):R107-R117. doi: 10.1530/EJE-20-0103. PMID: 32698138; PMCID: PMC7682746].

The Authos should for example put these 3 main genes’ descriptions’ on top and then put the others following.

22)     line 195, while citing that activating mutations in HPG system could potentially lead to CPP it is worth also citing recent article by Raja Brauner, Joelle Bignon-Topalovic, Anu Bashamboo  and Ken McElreavey , Frontiers in Pediatric 2021, where activating mutations in genes associated to HH, could potentially lead to peripheric PP from ovaric cysts.

Author Response

Dear Reviewer,

Thank you for your helpful comments. After taking into account the propositions you have made, our responses are as follows:

Point number 1: 

The most common genes known to be associated in the conditions (MRKNS, DLK1 And KISS-KISS1R) should be stressed further.

Mutation in KISS1-KISS1 r should be highlighted as gain of function, mutation in MKRN3  protein that physiologically has a role in repressing pubertal  initiation should be highlighted as loss of function, while mutations in DLK1 gene, a paternally expressed gene very common in the adrenal, pituitary and ovaric tissue, should be described as loss of function.

[see reference: Roberts SA, Kaiser UB. GENETICS IN ENDOCRINOLOGY: Genetic etiologies of central precocious puberty and the role of imprinted genes. Eur J Endocrinol. 2020 Oct;183(4):R107-R117. doi: 10.1530/EJE-20-0103. PMID: 32698138; PMCID: PMC7682746].

The Authos should for example put these 3 main genes’ descriptions’ on top and then put the others following.

Answer number 1: More details have been provided. Gain and Loss of function mutations in these genes consequences have been explained in the discussion section. The proposed reference by Roberts et al. has now been cited in the text. These genes have been put more closely, with the sole interference of the PLCB1, PRKCA, ITPR1 genes, since they complete the KISSQ/KISS1R pathway.

Point number 2: 

 line 195, while citing that activating mutations in HPG system could potentially lead to CPP it is worth also citing recent article by Raja Brauner, Joelle Bignon-Topalovic, Anu Bashamboo  and Ken McElreavey , Frontiers in Pediatric 2021, where activating mutations in genes associated to HH, could potentially lead to peripheric PP from ovaric cysts.

Answer number 2:

The proposed article has been sited. 

We would like to thank you for your proposals and the interesting bibliography you recommended. Please see the attachment in order to access the newer version of the article.

Thank you for your time,

Kind regards,

KT

Reviewer 2 Report

The authors conducted a systematic review to investigate the role of various genes' single nucleotide polymorphisms (SNPs) in the pathogenesis of idiopathic central precocious puberty in girls. The topic is current, and since the exact mechanism underlying central precocious puberty remains unknown, systematic reviews and meta-analyses of genetic imprinting and the genetic basis are necessary. However, this manuscript requires several crucial improvements. 

The section that needs to be revised is Materials and methods. Authors need to provide a detailed research strategy. Database search needs to be expanded to EMBASE and Web of Science. How many reviewers independently reviewed all titles and abstracts and selected the potentially relevant publications? Did the authors investigate which diagnostic criteria were listed in the paper or provided by the study authors? The authors also need to provide primary (or other) outcome(s). What were the exclusion criteria for selected papers? 

Moreover, there are technical aspects of the manuscript that require revision. In the text, reference numbers should be placed in square brackets [ ], and placed before the punctuation. The reference list is not in accordance with the journal's propositions. It seems that the authors used the citation directly from PubMed. I encourage the authors to use Reference manager software (Zotero or EndNote) which includes the style files for the selected journal.  

The conclusion section needs to be separated from the Discussion. 

Lastly, there are some specific comments:

  1. There are parts of the Introduction that require more references - the first and fourth paragraphs. 
  2. The fifth paragraph in the Introduction section (regarding the importance of SNPs) should provide more details and examples of the general role of SNPs in various diseases and conditions. 

Author Response

Dear Reviewer,

First of all, we would like to thank you for your time and propositions concerning our paper. After receiving and taking into account your recommendations here are our answers.

Point number 1:

The section that needs to be revised is Materials and methods. Authors need to provide a detailed research strategy. Database search needs to be expanded to EMBASE and Web of Science. How many reviewers independently reviewed all titles and abstracts and selected the potentially relevant publications? Did the authors investigate which diagnostic criteria were listed in the paper or provided by the study authors? The authors also need to provide primary (or other) outcome(s). What were the exclusion criteria for selected papers? 

Answer number 1: The Materials and Methods section has been revised. A more extensive description of the strategy that was followed has been provided. We expanded our research to EMBASE and Google scholar and added 4 more articles to our review. We described the outcomes and exclusion criteria of our research.

Point number 2:

Moreover, there are technical aspects of the manuscript that require revision. In the text, reference numbers should be placed in square brackets [ ], and placed before the punctuation. The reference list is not in accordance with the journal's propositions. It seems that the authors used the citation directly from PubMed. I encourage the authors to use Reference manager software (Zotero or EndNote) which includes the style files for the selected journal.  

The conclusion section needs to be separated from the Discussion. 

Answer number 2: 

Reference numbers morphology has been updated. We used the Zotero platform in order to update references morphology. Conclusion was separated from the Discussion section.

Point number 3: 

  1. There are parts of the Introduction that require more references - the first and fourth paragraphs. 

Answer number 3: We have added more details that were necessary in these paragraphs

Point umber 4:

  1. The fifth paragraph in the Introduction section (regarding the importance of SNPs) should provide more details and examples of the general role of SNPs in various diseases and conditions. 

Answer number 4:

A more thorough research of literature concerning the role and importance of SNPs in humans was performed, adding more details in our manuscript.

Please find attached below our manuscript, after the suggested changes.

Thank you for your time,

Kind regards,

KT

Round 2

Reviewer 2 Report

The authors significantly improved the manuscript with the acceptance of previously stated suggestions.

There are only two points that need to be considered:

1. Materials and Methods, lines 101 and 102: Instead of full names, the authors should use the initials (K.T. and G.P.)

2. References

I still think the reference list is not following the journal's instructions. References should be described as follows:

Author 1, A.B.; Author 2, C.D. Title of the article. Abbreviated Journal Name Year, Volume, page range.

Here is the link for the Zootero software: https://www.zotero.org/styles/?q=id%3Amultidisciplinary-digital-publishing-institute 

The reference list in the manuscript does not have an abbreviated journal name. Moreover, the authors should not write "vol. number, Month, and Year". For example, reference number 42 should be written as:

Lima, C. J. G., et al. Mutational Analysis of the Genes Encoding RFAmide-Related Peptide-3, the Human Orthologue of Gonadotrophin-Inhibitory Hormone, and Its Receptor (GPR147) in Patients with Gonadotrophin-Releasing Hormone-Dependent Pubertal Disorders. J. Neuroendocrinol2014, 26, 817-24. https://doi.org/10.1111/jne.12207. 

Author Response

Dear Reviewer,

We would like to thank you for your comments and suggestions. We have updated our manuscript in accordance to your suggestions. More precisely,

Suggestion number 1. Materials and Methods, lines 101 and 102: Instead of full names, the authors should use the initials (K.T. and G.P.)

Answer number 1. We have replaced our full names with the abbreviated versions

Suggestion number 2. 

2. References

I still think the reference list is not following the journal's instructions. References should be described as follows:

Author 1, A.B.; Author 2, C.D. Title of the article. Abbreviated Journal Name Year, Volume, page range.

Here is the link for the Zootero software: https://www.zotero.org/styles/?q=id%3Amultidisciplinary-digital-publishing-institute 

The reference list in the manuscript does not have an abbreviated journal name. Moreover, the authors should not write "vol. number, Month, and Year". For example, reference number 42 should be written as:

Lima, C. J. G., et al. Mutational Analysis of the Genes Encoding RFAmide-Related Peptide-3, the Human Orthologue of Gonadotrophin-Inhibitory Hormone, and Its Receptor (GPR147) in Patients with Gonadotrophin-Releasing Hormone-Dependent Pubertal Disorders. J. Neuroendocrinol2014, 26, 817-24. https://doi.org/10.1111/jne.12207. 

Answer number 2. We have used the Zotero application, after adaption for Children Journal. However, after rechecking some references, we realised that some Journals' titles were not abbreviated, even though they appeared as such on the applicarion, so we manually updated them, using the NLM catalog.

Please find attached below the updated-corrected version of our manuscript,

Kind regards,

Konstantina Toutoudaki
